# LoRe - Logarithm Regularization for Few-Shot Class Incremental Learning

## Abstract

Few-Shot Class-Incremental Learning (FSCIL) aims to adapt to new classes with very limited data, while remembering information about all the previously seen classes. Current FSCIL methods freeze the feature extractor in the incremental sessions to prevent *catastrophic forgetting*. However, to perform well on the incremental classes, many methods reserve feature spaces during base training to allow sufficient space for incremental classes. We hypothesize that such feature space reservation sharpens the minima of the loss-landscape, resulting in sub-optimal performance. Motivated by the superior generalization of wide minima, we propose *LoRe* - logarithm regularization to guide the model optimization to wider minima. Moreover, we propose a denoised distance metric when considering similarity with the poorly calibrated prototypes. Comprehensive evaluations across three benchmark datasets reveal that *LoRe* not only achieves state-of-the-art performance but also produces more robust prototypes. Additionally, we demonstrate that *LoRe* can be leveraged to enhance the performance of existing methods.

## 1 Introduction

Modern artificial intelligence models, much like humans, are expected to continuously adapt and learn. However, traditional deep learning approaches typically require large datasets to achieve optimal performance. For instance, in class-incremental learning (CIL) Hou et al. (2019); Li & Hoiem (2018); Rebuffi et al. (2017); Masana et al. (2022), models depend on substantial amounts of data arriving in incremental sessions to adapt to new tasks. This reliance on extensive datasets for each incremental task is often unrealistic. For example, a voice recognition system should be able to identify new voices without needing hours or days of speech data for each new voice. As a result, Few-Shot Class Incremental Learning (FSCIL) methods Zhang et al. (2021); Peng et al. (2022); Song et al. (2023); Zhou et al. (2023; 2022) have garnered significant attention in recent years. In an FSCIL framework, ample data is available only for the base classes, while the incremental sessions provide only a limited number of examples for new classes.

The primary challenge in continual learning is achieving a balance between stability and plasticity, i.e. retaining previously learned information while adapting to new data. This problem is further exacerbated in FSCIL settings, where only a limited number of data points are available for new classes. Consequently, incrementally-trained models are prone to overfitting on the new data, thereby leading to *catastrophic forgetting* Rebuffi et al. (2017); Castro et al. (2018); Tao et al. (2020). To mitigate this issue, many recent FSCIL methods Wang et al. (2023); Peng et al. (2022); Zhou et al. (2022); Song et al. (2023) limit training to the base session. During the incremental sessions, the backbone of the model is frozen, preventing updates that could result in catastrophic forgetting, and it is only utilized to encode the data from the new classes.

Since training is confined to the base session, it is essential to adapt the training approach to accommodate new classes. Recent studies indicate that using cross-entropy loss can be sub-optimal for effectively separating representations Peng et al. (2022); Song et al. (2023). Consequently, alternate methods have been introduced for base-training. Peng et al. (2022) introduce a large-margin angular penalty that minimizes intra-class distance while maximizing inter-class separation. Song et al. (2023) propose a semantic-aware virtual contrastive loss, which incorporates "fantasy" classes (created by augmenting base classes) into the base training. This technique helps cluster representations of the base classes together, creating more space in the feature space for novel (incremental) classes.

Zhou et al. (2022) intentionally reserve areas in the feature space during base training by employing virtual prototypes, ensuring that adequate space is available for accommodating incremental classes. We hypothesize that artificially constraining the feature space to ensure room for novel classes increases the sharpness of the loss minima. (A sharp minima is characterized by a rapid change in loss value in its vicinity, while a flat/wide minima exhibits a slow variation in loss value nearby.). This is because a smaller portion of the loss landscape becomes optimal for these artificial tasks, leading increased sensitivity to perturbations and steeper minima.

Many works have found wider minima to generalize better to unseen data Keskar et al. (2017); Izmailov et al. (2019); Foret et al. (2021); Zhang et al. (2024); Chaudhari et al. (2017). The reason for this is that wide-minima ensure optimal performance is cases of some shift between the training and testing loss surface Keskar et al. (2017). Consequently, flat minima are more robust to distribution shifts. However, finding wide minima is non-trivial. Izmailov et al. (2019) propose stochastic weight averaging (SWA) of model weights when trained with constant/cyclical learning rate. Foret et al. (2021) propose a sharpness-aware minimization (SAM) strategy which considers the local loss neighbourhood of a point, and drive optimization towards to large regions of low loss. However, they are incompatible in the FSCIL setting - SWA imposes constraints on the learning rate Zhang et al. (2024), whereas SAM Foret et al. (2021) does not take incremental classes into consideration.

To overcome this, we propose *LoRe*, logarithm regularization to inject information from a flattened loss-landscape during gradient calculation to promote convergence to wider minima. *LoRe* can be easily integrated into model training with minimal effort, thereby enabling us to incorporate *LoRe* within existing methods to obtain state-of-the-art performance. Moreover, we identify systematic differences between the prototypes of base classes and novel classes derived from existing methods. To address these differences, we introduce a denoised distance metric when evaluating the closest class prototype.

Specifically, we make the following contributions -

1) We propose *LoRe* - Logarithm Regularization to inject information from a widenend loss-landscape during model optimization to guide the model towards wider minima.

2) We uncover systematic differences between the base-class and novel-class prototypes obtained from existing methods. To overcome these, we propose a denoised distance metric when calculating the nearest class mean.

3) We benchmark our approach on 3 benchmark datasets and obtain state-of-the-art performance. Through experiments, we also show that *LoRe* improves the performance of existing methods.

4) Through experiments, we also show that regularizing a model with *LoRe* leads to representations which are more robust to noise.

## 2 RELATED WORK

### 2.1 FEW-SHOT CLASS INCREMENTAL LEARNING

Few-shot class-incremental learning methods aim to adapt to incremental classes with few data-points while remembering information from the base classes. Earlier works in the space aimed to adapt class-incremental learning methods into the few-shot regime. Castro et al. (2018); Rebuffi et al. (2017) jointly learn the data encoder and the classifier, using a combination of distillation, to retain already learnt knowledge and cross-entropy to learn new classes. Recent works in FSCIL focus on novel strategies for base-training to enhance separation of incremental classes. Zhang et al. (2021) first proposed decoupling the learning of the representations and classifiers, with only the classifiers being updated in the incremental sessions. Peng et al. (2022) introduce a large-margin angular penalty that minimizes intra-class distance while maximizing inter-class separation, thereby allowing sufficient space for incremental classes. Zhou et al. (2022) intentionally reserve areas in the feature space during base training by employing virtual prototypes and predicting new classes. Song et al. (2023) propose a semantic-aware virtual contrastive loss, which incorporates "fantasy" classes (created by augmenting base classes) into the base training. This technique helps cluster representations of the base classes together, creating more space in the feature space for novel (incremental)

classes. Wang et al. (2023) propose a training-free prototype-calibration mechanism which uses the well-calibrated base-class prototypes to calibrate the novel class prototypes.

## 2.2 WIDE MINIMA IN DEEP LEARNING

Several works have examined the generalization properties of wide minima. Keskar et al. (2017) demonstrated the superiority of wide minima by varying the batch sizes. They showed that large batch sizes converged to sharp minima and exhibited sub-optimal generalization. Small batch sizes, on the other hand, converged towards wide minima and demonstrated superior test performance. Wide minima have a large proportion of almost-zero eigen values - Chaudhari et al. (2017) leverage this observation to devise Entropy-SGD, an objective function that favors approximate solutions lying in flatter regions of the loss landscape (wide minima) and avoid solutions in sharp valleys (sharp minima). Izmailov et al. (2019) propose averaging of model weights along the training trajectory to converge to wide-minima. Foret et al. (2021) propose a sharpness-aware-minimization strategy which formulates model training as a min-max optimization problem by maximizing the neighbourhood size of uniform loss around a loss minima. Zhang et al. (2024) point out how stochastic weight averaging Izmailov et al. (2019) is sensitive to the learning rate used, and propose a Lookahead strategy which involve weight interpolation to ensure convergence. while these methods aim to find wide minima in a typical image classification setup, they do not take into account the incremental learning requirements. F2M SHI et al. (2021) aim to overcome catastrophic forgetting by finding flat minima of the base classes and fine-tuning within the region during the incremental sessions. While the method confirms our hypothesis of flat minima helping prevent catastrophic forgetting, it exhibits suboptimal performance because it uses the cross-entropy loss (which has been shown to be sub-optimal Peng et al. (2022); Song et al. (2023)) and does not use the latest advancements in computer vision (such as contrastive learning). Our method, *LoRe* is easily integrable within any framework, and can therefore be used to improve the performance of the latest methods.

## 3 METHOD

### 3.1 PROBLEM SETUP

The FSCIL problem setup consists of one base session with sufficient training data and multiple incremental sessions, each consisting of limited training data. The goal is to learn a model which is able to perform well on the tasks in the incremental sessions without *forgetting* about the base session task.

To be precise, FSCIL problems often assume $m+1$ sessions, with $\{D_0^{train}, D_1^{train}, ....D_m^{train}\}$ being the training set for each session, and $\{D_0^{test}, D_1^{test}, ....D_m^{test}\}$ being the corresponding testing sets. The $D_0^{train}/D_0^{test}$ represent the training/ testing data of the base session and $D_i^{train}/D_i^{test}, i \in [1...m]$ corresponds to incremental session data. Here, we consider a classification problem, where each $D_i^{train}/D_i^{test}$ consists of labelled image pairs $(x_k, y_k)$ and the task is learn to classify the images correctly. Typically, there is sufficient labelled data for the base session, i.e. $|D_0^{train}|$ is large, whereas each incremental session is typically an $N-way-K-shot$ classification task, where the goal is to learning to differentiate between $N$ classes, with only $K$ images available per class, following Vinyals et al. (2016); Tao et al. (2020). There is no overlap between the classes of any sessions (base/ incremental), i.e. if $C_i$ is the set of all classes seen in the $i_{th}$ session, then $C_i \cap C_j = \phi, if i \neq j, i, j \in [0...m]$. The training data is streaming in nature, i.e. $D_i^{train}$ is only seen by the model in the $i_{th}$ session and is not accessible in any other session $j \neq i$. On the other hand, the model, in the $i_{th}$ session, is tested on all classes seen so far, i.e. $C_0 \cup C_1.....C_i$. Typically, the testing set is balanced consisting of equal amounts of data from base and incremental classes alike. The goal of the model is to adapt well to the incremental classes, without forgetting information about the base classes.

Many previous works Peng et al. (2022); Zhang et al. (2021); SHI et al. (2021) adopt the incremental-frozen framework, where a classifier is learnt during the base session using the large amount of base data, with various provisions to accomodate the incremental classes. The classifier $\phi$ consists of 2 components - a feature extractor $f$ and a linear classification head $W$, i.e. for an input sample $x$, $\phi(x) = W^T f(x)$, where $phi(x) \in \mathbb{R}^{|C_0| \times 1}, f(x) \in \mathbb{R}^{d \times 1}$ and $W \in \mathbb{R}^{d \times |C_0|}$, where $d$ is the dimension of the feature extractor. In essence, $W$ consists of prototypes of the base classes in the

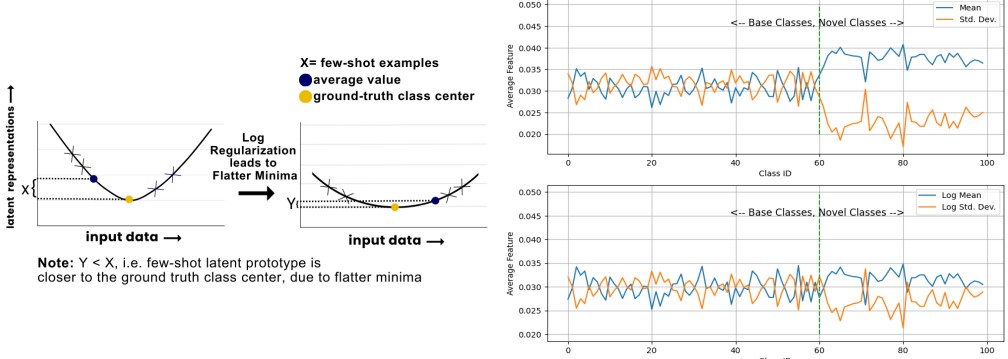

Figure 1: Motivation for the proposed methodology. Wider minima benefits few-shot class incremental learning by learning inherently robust prototypes (left). Using the logarithmic distance further boosts performance by denoising the prototypes (right)

base session. In the incremental sessions, the feature extractor is frozen and $W$ is expanded with the prototypes of novel classes. The final prediction is made in terms of the *nearest class mean* (NCM) Mensink et al. (2013) algorithm, calculated as

$$c_x = \operatorname{argmax}_i sim(f(x), w_i), \tag{1}$$

where $sim$ is the cosine similarity between two vectors.

## 3.2 Logarithm Regularization

Due to the problem setting of FSCIL, very few datapoints are available from the incremental classes. This inherently makes the incremental class prototypes poorly calibrated. Moreover, several methods incorporate constraints in the base training to reserve feature space for the incremental classes Zhou et al. (2022); Song et al. (2023). While it clusters the base class representations together enabling base class separation, it also constrains a large number of base classes into a smaller portion of the feature space. We hypothesize that that these constraints result in sharpening the minima in the loss landscape, since a smaller portion of the feature space is available of optimization on the *actual* base task. Models convergent on sharp minima are more susceptible to noise/ perturbations. Given that only few-shot examples are available from incremental classes, sharper minima compound the prototype-calibration problem, leading to sub-optimal performance. An example of this is shown in Fig. 1 (right) - the figure on the top shows the L1-norm (blue) and standard deviation (orange) of the *L2 - normalized* prototypes obtained from ALICE Peng et al. (2022) on the CIFAR-100 dataset. Evidently, the L1-norm for the incremental class prototypes is larger than that of the base classes with much smaller standard-deviation, indicating that many features in the incremental-class prototypes are of roughly similar value. This highlights the poor calibration of incremental class prototypes. Moreover, model convergence to sharp minima is more sensitive to hyperparameters such as the learning rate.

Wider minima have been demonstrated to have better generalization properties. Keskar et al. (2017); Izmailov et al. (2019); Foret et al. (2021). However, finding wide minima in large models is nontrivial. Methods such as Sharpness-Aware Minimization Foret et al. (2021) formulate model optimization as a min-max problem, maximizing the neighbourhood with uniformly low loss. However, its extension into FSCIL settings is non-trivial because finding wider neighbourhoods of minima during base-training might lead to increased plasticity of the model. Instead, we approach the problem by augmenting the loss-landscape, attempting to inject information from a wider loss-landscape to aid model optimization. We call it *LoRe* or logarithmic regularization. Specifically, if $L$ is the loss function (typically, cross-entropy for classification problems) used to optimize a neural network, we propose a regularized loss function $\hat{L}$, such that -

$$\hat{L} = L + \lambda * \frac{1}{|w|} \Sigma_{\forall w} log(1 + ||w||_2) \tag{2}$$

where $|w|$ is the total number of parameters in the network, $||.||_2$ denotes the L2-norm of the weights. $\lambda$ is a hyper-parameter - we tried out various values of $\lambda$ from 0.1 to 1e-6 and found a value of 1e-5 to work well on all datasets.

The $log$ function smoothens the loss landscape, widening the minima with respect to the weights the weights (see Fig. 2). We hypothesize that regularizing a model with *LoRe* helps guide the gradient with information from a widened loss landscape, thereby aiding convergence to flatter minima, and resulting in more robust representations.

Figure 2: Flattening the minima with the log function

### 3.3 DENOISED DISTANCE

As mentioned earlier, we observe systematic differences in the scale of the prototypes of the base and novel classes. Specifically, in Fig 1, we observe that the L1-norm of the incremental class prototypes (L2-normalized) is significantly larger than the base class prototypes. This biases the inner product calculation, due to the difference in scale. We attribute this to the strict constraints imposed during the base training to reserve feature space for the incremental classes. To overcome this, we attempt remove with this difference in scale, before calculating the inner product. Specifically, we propose a modified distance measure when comparing the similarity between two vectors. Specifically, for two vectors $x$ and $y$, we propose a logarithmic inner-product distance, as -

$$< x.y > = \frac{\hat{x}.\hat{y}}{||\hat{x}||||\hat{y}||}; \hat{x}, \hat{y} = log(1 + x), log(1 + y); \tag{3}$$

The $log$ function is a concave function, making it suitable for scaling the prototypes and representations. Moreover, the prototypes and representations are also often learnt using a *ReLU* function, thereby making them compatible with the $log$ function.

## 4 EXPERIMENT

### 4.1 DATASETS

Following the setting of Zhang et al. (2021), we evaluate *LoRe* on three benchmark datasets - CUB200 Wah et al. (2011), CIFAR100 Krizhevsky & et al. (2009) and miniImageNet Russakovsky et al. (2015). Their details are as follows -

- **CIFAR100** Krizhevsky & et al. (2009): The CIFAR100 dataset consists of 60,000 images from 100 classes. Each image has a size of 32 x 32 pixels. The 100 classes are split into 60 base classes and 40 incremental classes, consisting of eight 5-way-5-shot incremental sessions.
- **CUB200** Wah et al. (2011): The CUB200 dataset is a fine-grained classification dataset consisting of 11,788 images from 200 classes of birds. Each image has a size of 224 x 224 pixels. The 200 classes are split into 100 base classes and 100 incremental classes, consisting of ten 10-way-5-shot incremental sessions.
- **miniImageNet** Russakovsky et al. (2015): The miniImageNet dataset consists of 60,000 images from 100 classes. Each image has a size of 84 x 84 pixels. The 100 classes are split into 60 base classes and 40 incremental classes, consisting of eight 5-way-5-shot incremental sessions.

All methods are evaluated on the same train/test splits, following Zhang et al. (2021) to ensure a fair comparison.

## 4.2 EVALUATION

To evaluate the performance of various methods, we report the average accuracy and harmonic accuracy in each session. The average accuracy is the classwise accuracy, averaged over all the classes in the current session. However, due to the high proportion of base classes among the overall number of classes, it is possible to have a high average accuracy while performing poorly on the incremental classes. Hence, following Peng et al. (2022), we also report the harmonic accuracy in each session. The harmonic accuracy is calculated as the harmonic mean of the average base class accuracy and average incremental accuracy (averaged of all incremental classes seen upto the current session)

## 4.3 IMPLEMENTATION DETAILS

We adopt the ResNet18 He et al. (2016) architecture for experiments. Follwing previous work Zhou et al. (2022); Peng et al. (2022); Song et al. (2023), the model is trained from scratch for the miniImageNet and CIFAR100 datasets and initialized with the pretrained ImageNet weights for the CUB200 datasets. Since we propose a regularization method, we incorporate it within each method Wang et al. (2023); Peng et al. (2022); Song et al. (2023) and use their implementation details, without modification. For example, ALICE Peng et al. (2022) uses class- and data- augmentation for the CIFAR100 and miniImageNet datasets, but omits the class augmentation for the CUB200 dataset; we follow the same implementation setup. However, we train for an additional 30 epochs in each setup. For example, if ALICE is trained for 120 epochs, we train our *LoRe* model for 150 epochs. We adopt the same learning rate, augmentation and hyperparameter configurations as any original method, without making any modifications.[1]

## 5 RESULTS

### 5.1 COMPARISON WITH STATE-OF-THE-ART

We compare *LoRe* with several existing classical continual learning methods, namely iCARL Rebuffi et al. (2017), EEIL Castro et al. (2018) and TOPIC Tao et al. (2020) and state-of-the-art FSCIL methods namely CEC Zhang et al. (2021), FACT Zhou et al. (2022), LIMIT Zhou et al. (2023), TEEN Wang et al. (2023), ALICE Peng et al. (2022) and SAVC Song et al. (2023) on three benchmark datasets, namely CUB200, CIFAR100 and miniImageNet datasets. Table. 1 shows the detailed session-wise average accuracy comparison for all methods on the CUB200 dataset. We show that *LoRe*, when integrated with existing methods such as SAVC Song et al. (2023) and ALICE Peng et al. (2022), achieves state-of-the-art performance. SAVC Song et al. (2023), when optimized with *LoRe*, outperforms the existing state-of-the-art method by achieving **+1.43%** performance improvement in the final session accuracy and a **+1.87%** improvement in the average accuracy across sessions. Besides average accuracy, *LoRe* also achieves state-of-the-art performance in terms of harmonic accuracy on the CUB200 dataset, as shown in Table. 2. Fig. 3 shows the performance of various methods on the CUB200, CIFAR100 and miniImageNet datasets. *LoRe* achieves the highest final session accuracy on all datasets, outperforming the existing state-of-the-art method SAVC Song et al. (2023) by 1.43% , 3.72% and 2.23% respectively.

### 5.2 IMPROVING EXISTING METHODS

The proposed method, *LoRe*, is easily integrable within existing methods, and can be used to improve performance by guiding the optimization towards wider minima, which enable better calibrated class prototypes. When optimized with *LoRe*, the performance of ALICE Peng et al. (2022) improves from 58.70% to 59.89% in the final session on the CUB200 dataset, as shown in Table.1. To demonstrate this further, we incorporate *LoRe* within the learning framework of TEEN Wang et al. (2023), a training-free prototype calibration method which uses the well-calibrated base class prototypes to calibrate the novel class prototypes. TEEN uses a vanilla cross-entropy objective feature-extractor, making it an efficient and elegant solution for few-shot class incremental learning.

---

[1]The code for our implementation will be made publicly available.

| Method | Accuracy in each session(%) | | | | | | | | | | | Avg. | Δ |
|---|---|---|---|---|---|---|---|---|---|---|---|---|---|
| | 0 | 1 | 2 | 3 | 4 | 5 | 6 | 7 | 8 | 9 | 10 | | |
| FT-CNN* | 68.68 | 43.70 | 25.05 | 17.72 | 18.08 | 16.95 | 15.10 | 10.06 | 8.93 | 8.93 | 8.47 | 21.97 | +47.09 |
| iCARL* | 68.68 | 52.65 | 48.61 | 44.16 | 36.62 | 29.52 | 27.83 | 26.26 | 24.01 | 23.89 | 21.16 | 36.67 | +32.39 |
| EEIL* | 68.68 | 53.63 | 47.91 | 44.20 | 36.30 | 27.46 | 25.93 | 24.70 | 23.95 | 24.13 | 22.11 | 36.27 | +32.79 |
| TOPIC* | 68.68 | 62.49 | 54.81 | 49.99 | 45.25 | 41.40 | 38.35 | 35.36 | 32.22 | 28.31 | 26.28 | 43.92 | +25.14 |
| CEC | 76.32 | 71.88 | 67.04 | 62.24 | 61.30 | 57.38 | 56.04 | 54.29 | 52.57 | 51.32 | 49.84 | 60.02 | +9.04 |
| FACT | 75.89 | 73.34 | 70.20 | 65.21 | 64.67 | 61.49 | 60.73 | 59.31 | 57.69 | 57.22 | 56.20 | 63.81 | +5.25 |
| LIMIT | 79.66 | 76.52 | 73.05 | 68.09 | 67.50 | 63.54 | 62.51 | 61.43 | 60.19 | 58.99 | 57.50 | 66.27 | +2.79 |
| TEEN | 79.33 | 75.23 | 71.79 | 67.07 | 66.43 | 63.25 | 61.74 | 60.89 | 59.46 | 58.70 | 57.92 | 65.62 | +3.44 |
| ALICE | 72.80 | 70.41 | 68.78 | 65.45 | 64.00 | 61.58 | 60.90 | 60.01 | 58.87 | 59.10 | 58.70 | 63.69 | +5.37 |
| SAVC | 78.63 | 75.53 | 71.71 | 69.56 | 67.82 | 65.19 | 64.20 | 63.10 | 61.45 | 61.25 | 60.65 | 67.19 | +1.87 |
| ALICE + LoRe (ours) | 77.20 | 74.10 | 71.87 | 68.29 | 66.48 | 63.78 | 62.92 | 62.02 | 60.62 | 60.47 | 59.89 | 66.15 | |
| **SAVC + LoRe (ours)** | **80.48** | **77.38** | **74.75** | **71.19** | **69.76** | **66.94** | **65.97** | **65.08** | **63.22** | **62.81** | **62.08** | **69.06** | |

Table 1: Detailed session-wise accuracy of *LoRe* and baselines on the CUB200 dataset. *LoRe* combined with SAVC Song et al. (2023) produces state-of-the-art performance. * = performance reported in prior works.

| Method | Harmonic Accuracy in each session (%) | | | | | | | | | | Avg. | Δ |
|---|---|---|---|---|---|---|---|---|---|---|---|---|
| | 1 | 2 | 3 | 4 | 5 | 6 | 7 | 8 | 9 | 10 | | |
| FACT | 62.07 | 58.17 | 50.57 | 52.38 | 50.54 | 51.44 | 51.96 | 50.44 | 51.85 | 51.67 | 53.11 | +6.87 |
| TEEN | 64.46 | 60.78 | 54.50 | 55.88 | 54.50 | 54.76 | 54.52 | 53.18 | 54.36 | 54.52 | 56.15 | +3.83 |
| ALICE | 67.78 | 64.78 | 58.13 | 58.23 | 56.06 | 57.08 | 57.10 | 56.43 | 57.59 | 57.72 | 59.09 | +0.89 |
| SAVC | 60.47 | 59.49 | 55.44 | 56.23 | 54.46 | 55.78 | 56.40 | 55.50 | 56.81 | 57.24 | 56.78 | +3.20 |
| SAVC + LoRe (ours) | 63.60 | 61.86 | 56.21 | 58.63 | 56.01 | 57.34 | 58.28 | 57.02 | 58.07 | 58.33 | 58.54 | |
| **ALICE + LoRe (ours)** | **69.76** | **65.93** | **59.08** | **58.98** | **56.66** | **57.79** | **58.04** | **57.16** | **58.13** | **58.23** | **59.98** | |

Table 2: Detailed session-wise *harmonic* accuracy of *LoRe* and baselines on the CUB200 dataset. *LoRe* combined with ALICE Peng et al. (2022) produces state-of-the-art performance. Due to space constraints, the lower performing methods have been omitted from the table.

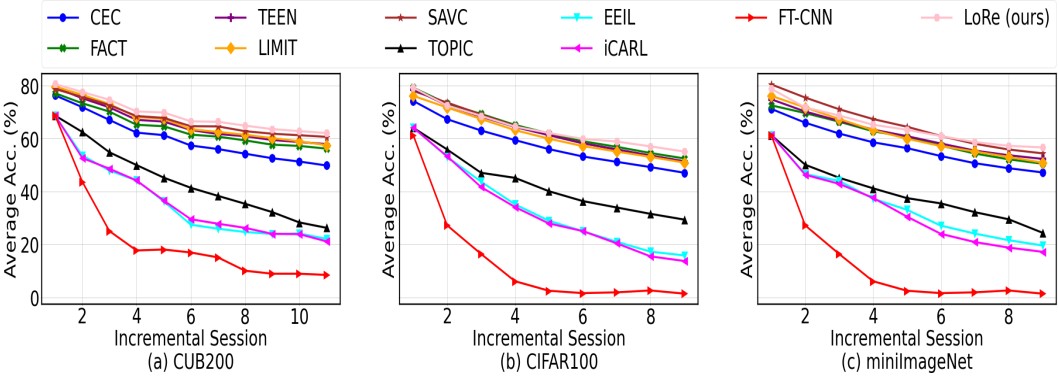

Figure 3: Comparision of performances of state-of-the-art methods with *LoRe*. *LoRe* outperforms SAVC Song et al. (2023) by 1.43% on the CUB200 dataset, 2.23% on the miniImageNet dataset and 3.72% on the CIFAR100 dataset in the last session accuracy. Detailed accuracies on the datasets can be found in the Supplementary Material.

| Method | Average Accuracy in each incremental session | | | | | | | | | | | Avg ↑ | Δ |
|---|---|---|---|---|---|---|---|---|---|---|---|---|---|
| | 0 | 1 | 2 | 3 | 4 | 5 | 6 | 7 | 8 | 9 | 10 | | |
| TEEN | 79.33 | 75.23 | 71.79 | 67.07 | 66.43 | 63.25 | 61.74 | 60.89 | 59.46 | 58.70 | 57.92 | 65.62 | +1.91 |
| TEEN+LoRe | 79.29 | 76.05 | 72.75 | 68.20 | 68.16 | 65.20 | 65.79 | 63.58 | 62.01 | 61.34 | 60.45 | 67.53 | - |
| Diff. | -0.04 | +0.82 | +0.96 | +1.13 | +1.73 | +1.95 | +4.05 | +2.69 | +2.55 | +2.64 | +2.53 | | |

Table 3: Performance of TEEN Wang et al. (2023) without/with LoRe on the CUB200 dataset. Addition of LoRe leads to a 2.53% improvement in the final session accuracy, and an average improvement of 1.91% across sessions.

| Method | Harmonic Accuracy in each incremental session | | | | | | | | | | Avg. ↑ | Δ |
|---|---|---|---|---|---|---|---|---|---|---|---|---|
| | 1 | 2 | 3 | 4 | 5 | 6 | 7 | 8 | 9 | 10 | | |
| TEEN | 64.46 | 60.78 | 54.50 | 55.88 | 54.50 | 54.76 | 54.52 | 53.18 | 54.36 | 54.52 | 56.15 | **+2.72** |
| TEEN + LoRe | **65.93** | **63.03** | **56.56** | **58.58** | **57.19** | **58.06** | **57.81** | **56.43** | **57.59** | **57.54** | **58.87** | - |
| Diff. | +1.29 | +2.25 | +2.06 | +2.70 | +2.69 | +3.30 | +3.29 | +3.25 | +3.23 | +3.02 | | |

Table 4: Performance of TEEN Wang et al. (2023) without/with LoRe on the CUB200 dataset. Addition of LoRe leads to a 3.02% improvement in the final session harmonic accuracy, and an average improvement of 2.72% across sessions in harmonic accuracy.

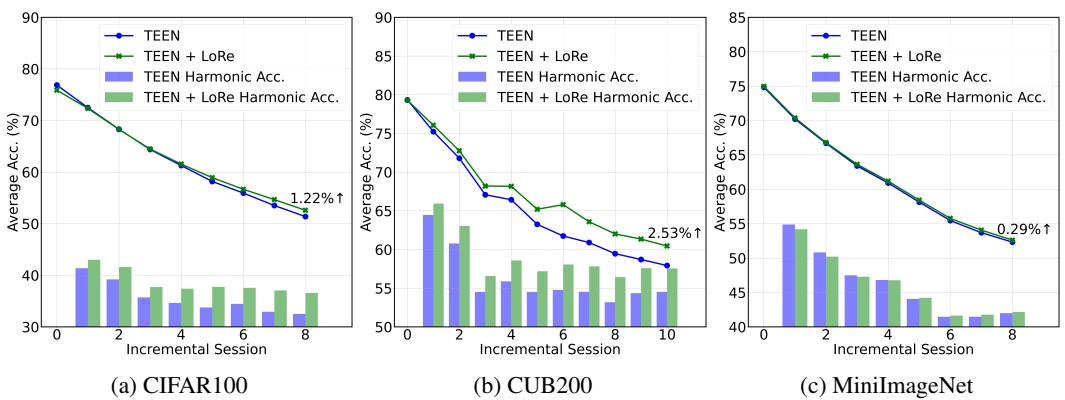

(a) CIFAR100          (b) CUB200          (c) MiniImageNet

Figure 4: Improvement in the average (and harmonic) accuracy of TEEN Wang et al. (2023) due to the addition of *LoRe*. Detailed accuracies on the datasets can be found in the Supplementary Material.

Table 3 shows the improvement in the performance of TEEN Wang et al. (2023) on the CUB200 dataset when optimized with *LoRe*. Addition of *LoRe* results in a **+2.53%** improvement in the final session average accuracy and an average increase of **+1.91%** across sessions. Moreover, the improvement in the average accuracy is due to improved performance on the incremental classes as demonstrated by the **+3.02%** increase in the final session harmonic accuracy and the **+2.72%** increase in the average harmonic across sessions in Table 4. Fig. 4 shows the improvement in the average (and harmonic) accuracy at each session on the CIFAR100, CUB200 and miniImageNet datasets, where the final session average ( harmonic) accuracy has improved by 1.22% (4.07%), 2.53% (3.02%) and 0.29% (0.18%), respectively. Addition of *LoRe* leads to improved performance of TEEN Wang et al. (2023) by guiding the optimization towards flatter minima.

## 5.3 ROBUSTNESS ANALYSIS

Since wider minima reduce the sensitivity to perturbations, prototypes learnt with *LoRe* must be more robust to noise. To verify this, we perform perturbation analysis wherein we manually perturb the prototypes from their original configuration and observe the effect on average and harmonic accuracy. Our expectation is that prototypes learnt with *LoRe* would be more robust to noise. To this end, we add random noise sampled from a uniform distribution, $U(0, \alpha)$, to the prototype, before calculating the cosine similarity. We vary $\alpha$ from 0 to 0.1 and observe the change in performance.

| Noise Level | 0 | 0.001 | 0.01 | 0.025 | 0.05 | 0.075 | 0.1 |
|---|---|---|---|---|---|---|---|
| SAVC | 60.67 (57.24) | 60.56 (57.13) | 60.75 (57.47) | 60.41 (57.12) | 59.16 (55.58) | 56.96 (53.08) | 54.35 (49.98) |
| SAVC + LoRe | **62.08 (58.33)** | **61.58 (57.96)** | **61.53 (57.88)** | **61.32 (57.54)** | **60.17 (56.00)** | **58.09 (53.51)** | **55.85 (51.01)** |
| Diff. | +1.41 (+1.09) | +1.02 ( +0.83) | +0.83 (+0.41) | +0.91 (+0.42) | +1.01 (+0.42) | +1.13 (+0.43) | +1.5 (+1.03) |

Table 5: Perturbation Analysis of SAVC Song et al. (2023) prototypes learnt without/with *LoRe* on the CUB200 dataset. *LoRe* prototypes are more robust to noise, as compared to the original SAVC prototypes and show higher average (harmonic) accuracy across sessions.

| | | Average Accuracy | | | | | | | | | | | | |
|---|---|---|---|---|---|---|---|---|---|---|---|---|---|---|
| LR | DD | 0 | 1 | 2 | 3 | 4 | 5 | 6 | 7 | 8 | 9 | 10 | Avg. | Δ |
| ✗ | ✗ | 78.63 | 75.53 | 71.71 | 69.56 | 67.82 | 65.19 | 64.20 | 63.10 | 61.45 | 61.25 | 60.65 | 67.19 | - |
| ✗ | ✓ | 78.39 | 75.22 | 72.74 | 69.61 | 68.00 | 65.49 | 64.43 | 63.47 | 61.83 | 61.65 | 61.06 | 67.40 | +0.21 |
| ✓ | ✗ | 80.27 | 77.25 | 74.52 | 70.87 | 69.19 | 66.46 | 65.45 | 64.67 | 62.85 | 62.47 | 61.68 | 68.70 | +1.51 |
| ✓ | ✓ | **80.48** | **77.38** | **74.75** | **71.20** | **69.76** | **66.94** | **65.97** | **65.08** | **63.12** | **62.81** | **62.08** | **69.05** | **+1.86** |

Table 6: Ablation Study of the average accuracy of *LoRe*, combined with SAVC Song et al. (2023). LR = Logarithmic Regularization and DD = Denoised Distance

Table 5 shows the performance of SAVC Song et al. (2023) prototypes learnt with and without *LoRe* with varying amount of noise on the CUB200 dataset. Prototypes learnt *LoRe* are more robust to noise and exhibit superior performance to the original method.

## 5.4 ABLATION STUDY

| | | Harmonic Accuracy | | | | | | | | | | | |
|---|---|---|---|---|---|---|---|---|---|---|---|---|---|
| LR | DD | 1 | 2 | 3 | 4 | 5 | 6 | 7 | 8 | 9 | 10 | Avg. | Δ |
| ✗ | ✗ | 60.47 | 59.50 | 55.44 | 56.62 | 54.46 | 55.78 | 56.40 | 55.50 | 56.81 | 57.24 | 56.82 | - |
| ✗ | ✓ | 58.90 | 59.64 | 55.76 | 57.10 | 55.01 | 56.02 | 56.80 | 55.87 | 57.23 | 57.67 | 57.00 | +0.82 |
| ✓ | ✗ | **64.47** | 61.78 | 55.81 | 57.74 | 55.50 | 56.87 | 58.04 | 56.86 | 57.86 | 58.07 | 58.30 | +1.48 |
| ✓ | ✓ | 63.60 | **61.86** | **56.21** | **58.63** | **56.01** | **57.34** | **58.28** | **57.02** | **58.07** | **58.33** | **58.54** | **+1.72** |

Table 7: Ablation Study of the harmonic accuracy of *LoRe*, combined with SAVC Song et al. (2023). LR = Logarithmic Regularization and DD = Denoised Distance

Tables 6 and 7 show an ablation study of *LoRe* using SAVC Song et al. (2023) as the base method, on the CUB200 dataset for average and harmonic accuracies, respectively. It must be noted that both components, logarithmic regularization and denoised distance, are essential in obtaining the observed improvement in performance. However, it can be said that the Logarithm Regularization is more important because it guides the optimization of the feature encoder, and determines what the prototypes look like, thereby showing a larger gain in performance. The logarithm distance, on the other hand, is a post-hoc method to denoise the distances when considering proximity to class prototypes, hence, resulting in a relatively smaller gain in performance. Nevertheless, both the components together help in achieving state of the art performance on all benchmarks. It must be noted that the proposed changes, not only help with the average accuracy across sessions, but also help in improving the harmonic accuracy with enhanced performance on incremental classes.

## 6 CONCLUSION

In this paper, we propose *LoRe* - logarithm regularization for to utilize gradient information from a flattened loss landscape to guide the model optimization towards wider minima. Further, we identify systematic differences between the base class and incremental class prototypes derived from existing methods and propose a denoised distance metric to overcome the bias. Evaluations across three benchmark datasets demonstrated that *LoRe* achieves state-of-the-art performance. Furthermore, *LoRe* can be seamlessly integrated into existing frameworks, and our results indicate that models trained with *LoRe* significantly outperform those that do not incorporate this approach.

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
