| Method | 0 | 1 | 2 | 3 | 4 | 5 | 6 | 7 | 8 |
|---|---|---|---|---|---|---|---|---|---|
| FT-CNN | 61.30 | 27.20 | 16.40 | 6.10 | 2.50 | 1.60 | 1.90 | 2.60 | 1.40 |
| iCARL | 64.10 | 53.28 | 41.69 | 34.13 | 27.93 | 25.06 | 20.41 | 15.48 | 13.73 |
| EEIL | 64.10 | 53.11 | 43.71 | 35.15 | 28.96 | 24.98 | 21.01 | 17.26 | 15.85 |
| TOPIC | 64.10 | 55.88 | 47.07 | 45.16 | 40.11 | 36.38 | 33.96 | 31.55 | 29.37 |
| SAVC | 78.02 | 73.60 | 69.09 | 64.77 | 61.41 | 58.28 | 55.79 | 53.63 | 51.37 |
| FACT | 77.67 | 73.20 | 69.31 | 65.15 | 61.94 | 58.99 | 56.88 | 54.54 | 52.47 |
| ALICE | 79.42 | 71.97 | 67.54 | 63.69 | 61.25 | 58.86 | 57.52 | 55.95 | 54.26 |
| CEC | 74.23 | 67.32 | 63.04 | 59.43 | 56.01 | 53.24 | 51.21 | 49.21 | 47.07 |
| LIMIT | 76.15 | 71.74 | 67.40 | 63.19 | 59.98 | 57.16 | 55.11 | 53.06 | 50.90 |
| TEEN | 76.87 | 72.48 | 68.31 | 64.39 | 61.28 | 58.19 | 55.94 | 53.52 | 51.38 |
| TEEN + CLOG | 76.80 | 72.34 | 68.27 | 64.48 | 61.55 | 58.93 | 56.67 | 54.66 | 52.60 |
| ALICE + LoRe | 79.23 | 72.48 | 68.59 | 64.52 | 62.11 | 59.87 | 58.88 | 57.03 | 55.09 |
| SAVC + LoRe | 77.95 | 73.65 | 69.17 | 64.99 | 61.63 | 58.75 | 56.51 | 54.37 | 51.93 |

Table 8: Detailed session-wise accuracy of various methods on the CIFAR-100 dataset

| Method | 0 | 1 | 2 | 3 | 4 | 5 | 6 | 7 | 8 |
|---|---|---|---|---|---|---|---|---|---|
| FT-CNN | 61.30 | 27.20 | 16.40 | 6.10 | 2.50 | 1.60 | 1.90 | 2.60 | 1.40 |
| iCARL | 61.30 | 46.30 | 42.90 | 37.60 | 30.50 | 24.00 | 20.90 | 18.80 | 17.20 |
| EEIL | 61.30 | 46.60 | 44.00 | 37.30 | 33.10 | 27.10 | 24.10 | 21.60 | 19.60 |
| TOPIC | 61.30 | 50.10 | 45.20 | 41.20 | 37.50 | 35.50 | 32.20 | 29.50 | 24.40 |
| CEC | 71.20 | 66.00 | 61.90 | 58.60 | 56.40 | 53.40 | 50.70 | 48.80 | 47.20 |
| FACT | 72.56 | 69.63 | 66.38 | 62.77 | 60.60 | 57.33 | 54.34 | 52.16 | 50.49 |
| TEEN | 74.83 | 70.22 | 66.70 | 63.40 | 60.91 | 58.14 | 55.46 | 53.70 | 52.32 |
| SAVC | 80.47 | 75.49 | 71.09 | 67.37 | 64.48 | 60.98 | 58.02 | 55.86 | 54.42 |
| ALICE | 79.33 | 71.06 | 67.86 | 64.32 | 62.14 | 59.72 | 57.41 | 56.38 | 55.44 |
| LoRe + ALICE | 78.78 | 71.77 | 68.73 | 65.23 | 63.33 | 60.95 | 58.68 | 57.26 | 56.65 |

Table 9: Detailed session wise accuracy on the miniImageNet dataset.

# A APPENDIX

## A.1 DETAILED SESSION WISE ACCURACY

Tables. 8 and 9 show the detailed session-wise accuracy of various methods on the CIFAR-100 and miniImageNet datasets.