# OpenReview forum: "LoRe - Logarithm Regularization for Few-Shot Class Incremental Learning"
_ICLR.cc/2025/Conference — ICLR 2025 Conference Withdrawn Submission_

### Official Review · Reviewer_nbTk · 2024-10-18

**Soundness:** 2
**Presentation:** 1
**Contribution:** 1
**Rating:** 3
**Confidence:** 5

**Summary:**

This paper argues that reserving feature space for new classes in a few-shot class incremental learning (FSCIL) scenario is harmful, as it leads to sharp minima, which result in poorer generalization in FSCIL. To address this, the authors propose a regularization term that encourages the model to converge to flat minima by regularizing the log of the L2 norm of the weights. Experimental results demonstrate that the proposed method can be applied to any existing FSCIL approach, improving performance.

**Strengths:**

This paper highlights that maintaining a large class margin among base classes to reserve space for new classes may be detrimental, a point with which the reviewer fully agrees.
The proposed denoised distance is somewhat valuable which might be easily overlooked.

**Weaknesses:**

The most significant weakness is the lack of novelty and justification. Inducing wide (or flat) minima was already proposed in F2M [1]. The authors claim that F2M achieves suboptimal performance because it does not leverage recent advancements in computer vision (L129-L130). However, this reasoning does not adequately justify the novelty or value of the proposed method compared to F2M.

Additionally, the justification for the proposed regularization is not well supported. It is described in lines 222-227, where the authors claim that regularizing the log of the L2 norm of weights 'guides the gradient with information from a widened loss landscape, aiding convergence to flatter minima.' However, the reviewer finds this explanation unclear and difficult to agree with, as there is no convincing evidence that the regularization leads to flatter minima. Moreover, the robustness analysis in Section 5.3 does not effectively demonstrate the method's effectiveness. If the proposed method truly identified flat minima, the performance difference is expected to increase with higher noise levels. Yet, as shown in Table 5, the performance difference does not increase across all noise levels, suggesting the difference stems merely from the model’s superior performance in the absence of noise.

In addition, the paper's presentation quality is poor. For example, Figure 2 lacks axis labels, making it difficult for readers to interpret. There are also many unnecessary notations in Section 3.1 that are not used later in the paper. Furthermore, several typos reduce the paper's overall quality (e.g., L222: 'the weights the weights', L144: 'D_0^t rain', L151: missing spaces and commas in the equation, etc.).

[1] Shi et al, "Overcoming catastrophic forgetting in incremental few-shot learning by finding flat minima." in NeurIPS 2021

**Questions:**

Why should we regularize the log of L2 norm of weights for finding flat minima?
More justification is required.

---

### Official Review · Reviewer_ZTYL · 2024-10-21

**Soundness:** 1
**Presentation:** 1
**Contribution:** 2
**Rating:** 3
**Confidence:** 5

**Summary:**

This paper proposes a logarithmic regularization method for few-shot class incremental learning (FSCIL). The authors hypothesize that existing FSCIL techniques that reserve feature space during base training lead to sharper loss minima. To address this, they try to guide model optimization towards wider minima by incorporating gradient information from a flattened loss landscape. The method also introduces a denoised distance metric to address systematic differences between base and novel class prototypes.

**Strengths:**

1. The paper proposes a novel regularization technique that can be integrated into existing FSCIL methods to improve performance.

2. Experimental results show improvements across multiple datasets and baselines when using LoRe.

**Weaknesses:**

1. The paper's overall quality is poor and requires substantial improvement across multiple aspects. The writing contains numerous issues, including inconsistent and incorrect use of mathematical symbols and formatting in the methodology section (e.g., the notation w in Equation 1 represents feature vector, however, in Equation 2, it represents the total parameter number; line 161 C0 represent the number of total class number, however, in 151, it represents the set of class). Citation formats do not adhere to conference standards. Font sizes for Figures 1 and 2 are too small, while Figure 3 lacks visual clarity. Several tables appear incomplete. These presentation issues significantly hinder the paper's readability and comprehension.

2. The paper's core motivation lacks sufficient evidence and fails to consider the broader landscape of FSCIL approaches. The authors hypothesize that existing space-saving FSCIL techniques increase loss minima sharpness, leading to poorer generalization. However, this claim is neither proven nor supported by empirical evidence or theoretical analysis. Furthermore, the paper neglects to address alternative FSCIL strategies such as those utilizing distribution shifts [1][2] or dynamic networks [3][4], which can achieve state-of-the-art performance without space preservation. This narrow focus undermines the proposed method's generalizability and the overall motivation of the work.

[1] Learnable Distribution Calibration for Few-Shot Class-Incremental Learning
[2] Few-Shot Class-Incremental Learning via Training-Free Prototype Calibration
[3] Exemplar-Based Contrastive Self-Supervised Learning with Few-Shot Class Incremental Learning
[4] MgSvF: Multi-Grained Slow versus Fast Framework for Few-Shot Class-Incremental Learning

3. The connection between poor prototype calibration and the sharp minima problem is not adequately established. The authors fail to provide convincing evidence that the observed calibration issues are directly caused by sharper loss landscapes. Additional empirical or theoretical analysis is necessary to support this aspect of the paper's argument. For example, theoretically, the authors can provide a mathematical derivation linking the concepts of loss landscape sharpness and prototype calibration, which can refer to generalization bounds and their relationship to loss landscape geometry.

4. The proposed logarithmic inner product distance is similar to cosine similarity, and the authors do not sufficiently differentiate their method. A thorough comparative analysis, including ablation studies or theoretical analysis, is needed to demonstrate the advantages of the proposed approach over existing similarity metrics. For example, the author can include ablation studies that utilize logarithmic distance compared to cosine similarity and other common metrics like Euclidean distance. The comparison should focus on multiple aspects, including computational efficiency, and performance on different types of datasets (fine-grained data, imbalance data, etc.).

5. The paper lacks a comprehensive description or visual representation of the proposed method's architecture. Without a clear overview of the network structure and final loss formulation, it is challenging to understand how the various components interact and function within the overall system. The overall visual diagram should illustrate the flow of data through the network, clearly showing how the feature extractor, classifier, and proposed regularization components interact. It should also depict how the logarithmic inner product distance is integrated into the classification process. Additionally, the authors should provide a clear mathematical formulation of the final loss function, showing how the various components (e.g., classification loss, regularization terms) are combined.

6. As the training setting is different compared with other benchmarks, more ablation studies should be conducted. For example, hyper-parameter sensitivity analysis, loss component analysis etc.

Minors

1. On line 144, the formatting for D^(train)_0 is incorrect. The use of absolute value symbols around D^(train)_0 on line 147 is unexplained. There is an inconsistency in the use of the symbol phi between lines 159 and 161.

2. The definition of the classifier on line 159 incorrectly includes the feature extractor, contradicting common terminology in the field.

3. In Equation 1, the meanings of variables c, x, and i are not clearly defined. Similarly, Equation 2 contains several confusing symbols that lack proper explanation.

**Questions:**

Given the significant weaknesses in methodology, presentation, and empirical validation, it seems that this paper does not meet the quality standards expected for publication at ICLR.

**Details Of Ethics Concerns:**

Nil

---

### Official Review · Reviewer_cMk5 · 2024-11-02

**Soundness:** 3
**Presentation:** 2
**Contribution:** 2
**Rating:** 5
**Confidence:** 5

**Summary:**

This paper introduced logarithm regularization into loss function to find a wider minima for model optimization, and proposed a denoised distance metric for classification. It achieves good performance on three benchmark datasets.

**Strengths:**

The motivation is clear and methodology is easy to follow, and the proposed method achieves good experiment results.

**Weaknesses:**

1. The key differences between this paper and F2M [1] need to be further clarified although the authors have mentioned F2M in "Related Work". It seems the motivation and solution of this paper is similar to F2M: they both searching for a flat minima for model optimization under few-shot incremental learning settings, and they both re-design the classification loss function to achieve this goal. Based on these, it seems that the novelty and contributions of this paper are limited.
2. The claim in L221: "The log function smoothens the loss landscape, widening the minima with respect to the weights the weights" lacks evidence. For instance, the log function may lead the model to converge to a suboptimal local point rather than the global flat minima. More theoretical or experimental evidence are needed to support this claim.
3. Line 245: "Moreover, the prototypes and representations are also often learnt using a ReLU function, thereby making them compatible with the log function", it would be better if the authors can give any explanation or evidence for this.
4. Many typos need to be fixed. For example, 1)Line 58: "nearby.)."; 2)Line 144:"D_0^{train}"; 3)Line 151:"ifi"; 4)Line 192:"that that".
[1] Overcoming Catastrophic Forgetting in Incremental Few-Shot Learning by Finding Flat Minima.

**Questions:**

Please kindly refer to the weaknesses, and here are some suggestions for the authors:
1. The authors could explicitly highlights any key technical differences in how to achieve flat minima, and how the proposed method advances the state-of-the-art beyond F2M's contributions.
2. The authors could provide some theoretical analysis of how the log function affects optimization dynamics, or more detailed visualizations of the loss landscape before and after applying the log function.
3. The authors could explain the mathematical or empirical relationship between ReLU and log functions that makes them compatible in this context, or provide some references or experimental results demonstrating this compatibility.

---

### Official Review · Reviewer_6k1z · 2024-11-02

**Soundness:** 2
**Presentation:** 2
**Contribution:** 2
**Rating:** 5
**Confidence:** 4

**Summary:**

The paper proposes a method called LoRe (Logarithm Regularization) for Few-Shot Class-Incremental Learning (FSCIL). The authors hypothesize that current methods that reserve feature spaces during base training to accommodate incremental classes lead to sub-optimal performance due to sharp minima in the loss landscape. To address this, they propose LoRe, which injects information from a wider loss landscape during model optimization to guide the model towards wider minima. They also introduce a denoised distance metric to address systematic differences between base class and incremental class prototypes. The proposed method is evaluated on three benchmark datasets and achieves state-of-the-art performance.

**Strengths:**

1) The proposed LoRe method is innovative and addresses the issue of sharp minima in the loss landscape, which is a common problem in FSCIL.
2) The addition of a denoised distance metric to address differences between base class and incremental class prototypes is a valuable contribution.
3) The paper is well-structured and provides a clear explanation of the proposed method, along with a thorough review of relevant literature.
4) The empirical findings demonstrate that LoRe achieves state-of-the-art performance and produces more robust prototypes.

**Weaknesses:**

1) The novelty of this manuscript is incremental. The authors applied Logarithm Regularization [1] into Few-Shot Class-Incremental Learning, which is already adopted in many methods, such as Numerical Simulation [2].
2) The motivation of this manuscript is not clear. The authors should clearly claim the challenging issues in previous methods, such as [3].
3) Some parts of the paper could be clearer in terms of exposition and explanation.
4) The authors complement the theoretical explanation of the success of the proposed approach.

[1] Regularized numerical methods for the logarithmic Schrodinger equation
[2] Regularization of the Logarithmic Mean for Robust Numerical Simulation
[3] Contrastive Augmented Graph2Graph Memory Interaction for Few Shot Continual Learning

**Questions:**

See Weaknesses

---

### Official Review · Reviewer_cJDW · 2024-11-04

**Soundness:** 3
**Presentation:** 3
**Contribution:** 2
**Rating:** 5
**Confidence:** 3

**Summary:**

The paper presents an approach called Logarithm Regularization (LoRe) for Few-Shot Class Incremental Learning (FSCIL). LoRe aims to achieve improved generalization and robustness by guiding the model optimization towards wider minima, which has been shown to generalize better under distribution shifts. The method introduces a denoised distance metric to handle calibration issues in prototypes for new classes. Evaluated on benchmark datasets such as CIFAR100, CUB200, and miniImageNet, LoRe demonstrates state-of-the-art performance when integrated with existing FSCIL frameworks.

**Strengths:**

1. **Innovation in Regularization:** The proposed Logarithm Regularization is a unique approach to guide models towards wide minima, addressing a known challenge in incremental learning, which is often prone to catastrophic forgetting and sensitivity to perturbations.

2. **Compatibility:** LoRe can be easily integrated with existing methods, offering performance improvements without significant modifications, making it a practical solution for enhancing current FSCIL techniques.

3. **Comprehensive Evaluation:** The paper includes thorough experiments across multiple datasets and metrics, including accuracy and harmonic accuracy, to demonstrate the robustness and generalization ability of LoRe.

**Weaknesses:**

1. **Lack of Theoretical Justification:** While the empirical results support the effectiveness of LoRe, a theoretical justification of why logarithmic regularization specifically leads to wider minima in FSCIL settings would strengthen the contribution.

2. **Incremental Improvement:** Although LoRe shows performance gains, the improvements are marginal in some cases. Additionally, Figure 3’s comparison may not be entirely accurate, as LoRe should be benchmarked on the same backbone as each baseline to ensure fairness. The results, when controlled for backbone, do not consistently show significant gains over SAVC.

3. **Denoised Distance Complexity:** The denoised distance metric, though beneficial, adds computational complexity. An analysis of its impact on model training time and computational resources would be helpful, especially for large-scale applications.

**Questions:**

Please refer to Weaknesses.

---

### Note · Authors · 2024-11-18

I have read and agree with the venue's withdrawal policy on behalf of myself and my co-authors.